# Host–Guest Interactions of Zirconium-Based Metal–Organic Framework with Ionic Liquid

**DOI:** 10.3390/molecules28062833

**Published:** 2023-03-21

**Authors:** Mohd. Faridzuan Majid, Hayyiratul Fatimah Mohd Zaid, Muhammad Fadhlullah Abd Shukur, Azizan Ahmad, Khairulazhar Jumbri

**Affiliations:** 1Department of Fundamental and Applied Sciences, Universiti Teknologi PETRONAS, Seri Iskandar 32610, Perak Darul Ridzuan, Malaysia; mohd._17006281@utp.edu.my (M.F.M.); mfadhlullah.ashukur@utp.edu.my (M.F.A.S.); khairulazhar.jumbri@utp.edu.my (K.J.); 2Centre of Innovative Nanostructures & Nanodevices (COINN), Universiti Teknologi PETRONAS, Seri Iskandar 32610, Perak Darul Ridzuan, Malaysia; 3Chemical Engineering Department, Universiti Teknologi PETRONAS, Seri Iskandar 32610, Perak Darul Ridzuan, Malaysia; 4Department of Chemical Sciences, Universiti Kebangsaan Malaysia, Bangi 43600, Selangor Darul Ehsan, Malaysia; azizan@ukm.edu.my; 5Department of Physics, Faculty of Science and Technology, Airlangga University (Campus C), Mulyorejo Road, Surabaya 60115, Indonesia; 6Centre for Research in Ionic Liquids (CORIL), Universiti Teknologi PETRONAS, Seri Iskandar 32610, Perak Darul Ridzuan, Malaysia

**Keywords:** metal–organic framework, ionic liquid, zeolitic imidazolate framework, density functional theory, structural parameters, intermolecular forces

## Abstract

A metal–organic framework (MOF) is a three-dimensional crystalline compound made from organic ligands and metals. The cross-linkage between organic ligands and metals creates a network of coordination polymers containing adjustable voids with a high total surface area. This special feature of MOF made it possible to form a host–guest interaction with small molecules, such as ionic liquid (IL), which can alter the phase behavior and improve the performance in battery applications. The molecular interactions of MOF and IL are, however, hard to understand due to the limited number of computational studies. In this study, the structural parameters of a zirconium-based metal–organic framework (UiO-66) and 1-ethyl-3-methylimidazolium bis(trifluoromethylsulfonyl)imide, [EMIM][TFSI] were investigated via a combined experimental and computational approach using the linker model approach. When IL was loaded, the bond length and bond angle of organic linkers were distorted due to the increased electron density surrounding the framework. The increase in molecular orbital energy after confining IL stabilized the structure of this hybrid system. The molecular interactions study revealed that the combination of UiO-66 and [EMIM][TFSI] could be a promising candidate as an electrolyte material in an energy storage system.

## 1. Introduction

A metal–organic framework (MOF) is a type of hierarchical nanomaterial that has nanoscale dimensions ranging from 1 to 100 nm. This material is made from metal clusters, which include copper, zinc, iron, aluminum and chromium, and is bridged via organic linkers, such as carboxylates, imidazolates, amines and pyridyls. The interconnectivity of these building blocks creates an array of polymer networks that contain voids with a high total surface area [1,2]. Some of the possible intermolecular interactions in an MOF are coordination bonds, electrostatic forces between charged metal and ligands, pi-stacking interactions via aromatic rings, van der Waals interactions between non-polar regions in an MOF and hydrogen bonding between an MOF and its neighbor molecules [3,4]. The size and shape of the pores in the MOF structure can be adjusted to specific applications, depending on the type of metal cluster and ligands used to synthesize the MOF [5,6,7]. These porous crystalline materials have a large total surface area (up to 6000 m^2^/g), pore volume (up to 90% free volume) and storage capacity.

On the other hand, ionic liquids (IL) are molten salt made from ions whose melting point is below some arbitrary temperature, such as 100 °C. Common cations for IL include imidazolium, ammonium, phosphonium and pyridinium, while anions are usually bromine, chlorine, bis(trifluoromethylsulfonyl)imide, tetrafluoroborate, hexafluorophosphate and trifluoromethanesulfonate. Some ILs can be made from two cations (dicationic IL), while the polymerization of IL can also lead to polymeric IL. The combination of different cations and anions may target specific applications, thus making IL a designer solvent. Unlike conventional organic solvents, IL has low vapor pressure, low flammability, exceptional thermal stability and high ionic conductivity [8,9].

Confining IL onto the micropores of an MOF is currently a research interest, as many scientists have reported improved performance in their niche applications [10,11,12,13]. The insertion of small molecules onto a host material can be described as a host–guest interaction, which is a non-covalent interaction between the guest molecule or ion and host molecule. This kind of interaction usually occurs in biological systems due to the presence of a cavity or binding site of macromolecules, such as protein–protein interactions, enzyme–substrate interactions, receptor–ligand interactions and DNA–protein interactions [14,15,16,17]. Host–guest interactions can occur through hydrophobic interactions, electrostatic forces, wan der Waals forces and hydrogen bonding, which eventually alter the behavior of the host material and guest molecules. Common effects of host–guest interactions are the control of the guest molecules’ orientation, the modification of the guest molecules’ reactivity and the encapsulation of the guest molecules inside the void of the host material.

The incorporation of IL into the micropores of MOF can alter the phase behavior of the hybrid material, which subsequently enhances the efficiency of the material in numerous applications, especially in energy storage systems. Few studies have reported the potential use of MOF@IL as an ion-conducting material in electrochemical studies. Dehdashtian, Pourreza and Rostamnia synthesized chromium-based MOF (MIL-101) loaded with 1-hexyl-3-methylimidazolium chloride as a working electrode for the development of an electrochemical sensor. When compared to a bare electrode, the proposed composite material has a high sensitivity for the detection of indole compounds, which is beneficial for efficient drug discovery [18]. MIL-101 also has beneficial use in battery systems. Su et al., synthesized MIL-101 with the incorporation of two types of IL, which were 1,3-dimethylimidazolium tetrafluoroborate and 1-Allyl-3-methylimidazolium dicyanamide. The ionic conductivity of the composite electrolyte can reach as high as 3.73 × 10^−3^ S cm^−1^, and it is regarded as a fast ionic conductor due to its low activation energy [19]. Zhang et al., fabricated surface-mounted HKUST-1 MOF loaded with 1-butyl-3-methylimidazolium bis(trifluoromethylsulfonyl)imide to study the measurement of ionic conductivity for energy storage applications. They compared the three different modes of electrolytes (excess IL, without excess IL and pellet). A ‘nanowetted interface’ was observed on the external surface of the MOF, promoting ionic conductivity; however, the intrinsic conductivities can only be measured without excess IL [20]. Wang et al. prepared a fuel cell membrane made of a zirconium-based MOF (UiO-66) with 1-butyl-3-methylimidazolium bis(trifluoromethylsulfonyl)imide as the plasticizer. The introduction of this composite material improved the performance of proton transfers (0.135 S cm^−1^ at 160 °C), which inspired the strategic design of a high-performance fuel cell system [21].

Despite the experimental efforts, there are limited publications on how the MOF and IL interact at a molecular level. It is difficult to observe the structural change in the confinement of IL inside the micropores of the MOF without employing a quantum approach. The molecular data will provide insights into the molecular design of a material as well as how to rationalize the experimental results. The first-principles calculations of MOF@IL allow the prediction of a material’s behavior via quantum mechanical considerations. The electronic structure of a material is calculated by solving Schrodinger’s equation with the proper selection of functionals. This can provide theoretical insights into the interaction of the MOF and IL, which can be supported by experimental evidence and eventually lead to the real application of properly designed MOF@IL hybrid materials.

In this study, structural characterization and first-principles calculations were performed to investigate the molecular interactions of MOF and IL. X-ray diffractometer (XRD), scanning electron microscopy (SEM), Fourier transformation infrared spectroscopy (FTIR) and Brunauer–Emmett–Teller (BET) Surface Area Analysis were used to study the structural properties of the MOF@IL hybrid system. Density functional theory calculation was used to optimize the model system and to obtain molecular insights based on the changes in structural parameters, electrostatic potential and molecular orbitals. A three-dimensional zirconium-based metal–organic framework (UiO-66) was selected as the model MOF and 1-ethyl-3-methylimidazolium bis(trifluoromethylsulfonyl)imide ([EMIM][TFSI]) was employed as the guest molecule. These materials are widely used in electrochemical systems [22,23,24,25,26,27,28].

## 2. Results and Discussion

### 2.1. Synthesis of UiO-66 and IL@UiO-66

In this study, UiO-66 crystals were synthesized via the solvothermal reaction of Zr^4+^ and terephthalic acid, mediated in DMF and modulated by acetic acid. To prevent any missing linkers during the synthesis of UiO-66, acetic acid was added to direct the formation of specific crystal faces and to enhance both particle size and porosity in UiO-66 [29]. The coordination of Zr^4+^ and terephthalic acid led to the self-assembly of metal–organic clusters. The clusters grew in size and formed nuclei, which made up the seed of the UiO-66 crystal. When more clusters were formed, the crystal grew in size and reached its final form. To form IL@UiO-66, IL was diffused into the pores of the UiO-66 structure via a wet-impregnation method using acetone, and it interacted with the metal nodes and organic linkers via host–guest interactions. Possible molecular interactions are coordination interactions of the electronegative atom from IL (nitrogen or oxygen) with the zirconium node and electrostatic forces between the cation and anion and weak van der Waals interaction.

### 2.2. Structural Characterizations

The powder form of UiO-66 was characterized with XRD to observe the crystallinity and reflection planes of the material. Figure 1 represents the diffractogram of as-synthesized UiO-66 and simulated UiO-66, which was calculated from the Crystallographic Information File (entry 4512072) generated via the Powder Diffraction Pattern utility in Vesta software. Similar XRD patterns can be seen from the two diffractograms, indicating the successful synthesis of UiO-66. The crystallizations of highly crystalline UiO-66 occur at cubic symmetry Fm3m space groups, similar to previously reported topology of UiO-66 [30,31,32,33]. Strong characteristic peaks were observed at 2θ = 7.47°, 8.63°, 12.17° and 25.85°, which corresponds to the parallel planes of (*h*, *k*, *l*) = (1 1 1), (0 0 2), (0 2 2) and (0 0 6), respectively. The small intensity on the reflection line at 12.17° was probably caused by the existence of diagonal linkers and terephthalate-zirconium brick interaction, which was responsible for the framework integrity of UiO-66. The highly intense harp peak at 7.47° and 8.63° corresponds to octahedral and tetrahedral cages, respectively [34].

The morphology of the surface of UiO-66 and IL@UiO-66 were characterized using Evo LS15 SEM. Because of the insulating properties of UiO-66, the samples were coated with a thin layer of gold first to reduce the charging of the surface to ensure a homogeneous analysis and imaging of the surface. The morphology of UiO-66 was visualized in Figure 2, while the different magnifications can be viewed in Appendix A. Larger particles of UiO-66 were observed as nanoparticles up to 2000 nm in size, which were similar to previous studies [35,36]. Homogenous particles of UiO-66 can be observed as cubic crystals, which have a cubic symmetry. Because of the limitation of SEM, the octahedral shape was hardly observed, which requires in-depth microscopy analysis. Figure 3, Appendix A represent the micrograph of the IL@UiO-66. The aggregation of [EMIM][TFSI] occurs on the surface of UiO-66, which alters the surface morphology of the framework structure. It was suspected that some molecules of [EMIM][TFSI] were incorporated inside the pores of UiO-66. The aggregation also reveals that some ILs could not penetrate the cages. According to Ferreira et al. (2019), small anions were preferentially located near the zirconium node, while cations and larger anions prefer to reside on the surface of UiO-66 due to the strong interactions with the organic linker, rather than at the cluster node site [37]. In another molecular dynamic simulation, Kanj et al. (2019) predicted that a bunching phenomenon may occur when impregnating a high amount of IL in MOF. The blockage of the MOF pores prevents the entry of IL, forcing them to be aggregated outside the pores [38]. Thus, the location of the incorporated ILs in UiO-66 was influenced by the charge, the size of IL and the loading of IL.

The energy dispersive spectrum was calculated for each sample during the micrograph analysis to identify the elemental composition of UiO-66 and IL@UiO-66. The atomic percentages of each element are presented in Figure 4 and Figure 5. For UiO-66, the carbon has the highest atomic percentage (66.86%), followed by oxygen (31.07%) and zirconium (2.07%). Meanwhile, when impregnating [EMIM][TFSI] inside UiO-66, all elements of ILs were present, indicating the insertion of guest molecules inside the host material.

The elemental imaging was performed to observe the distribution of the elements on the two materials. Figure 6 represents the mapping of carbon, oxygen and zirconium in UiO-66, while Figure 7 represents the mapping of carbon, oxygen, zirconium, fluorine, nitrogen and sulfur in IL@UiO-66. The elemental mapping demonstrated that the elements were distributed homogeneously in the structure.

To further understand the surface area of both materials, the physical adsorption of nitrogen was performed via the BET method. Table 1 shows the summary of the surface area and pore size of UiO-66 and IL@UiO-66, while their linear isotherms are illustrated in Figure 8 and Figure 9. The BET surface area of as-synthesized UiO-66 was 295.992 m^2^/g, which was similar to a previous report by Yin et al. [39]. The linear isotherm plot shows a Type I curve, implying the microporous behavior of UiO-66. When IL was loaded, the surface area decreased due to the dispersion of IL molecules on UiO-66, as depicted in the SEM image of IL@UiO-66. The linear isotherm plot for IL@UiO-66 can be classified as a Type II isotherm group due to the monolayer formation of IL aggregates [40].

FTIR characterizations were carried out to locate the functional groups and to elucidate the possible interactions between UiO-66 and [EMIM][TFSI]. Molecular interactions are verified via shifts in wavenumbers, changes in peak intensity and observations of the pattern of the FTIR spectrum. The FTIR spectrums of pristine UiO-66, IL and IL@UiO-66 were stacked for easier comparison and were divided into three different regions (Figure 10, Figure 11 and Figure 12). The OH region and framework nodes of UiO-66 can be identified in the wavenumber ranges of 3000–3600 and 500–1750 cm^−1^, respectively. Based on Figure 10, a broad OH stretch was clearly observed at 3300 cm^−1^, which refers to the condensation of the physiosorbed water inside the UiO-66 cage. A small OH peak at 3648 cm^−1^ was assigned to the small fraction of the isolated hydroxyl group on the external surface of crystal UiO-66. Noticeable peaks at 2979 and 2931 cm^−1^ were assigned to aliphatic and aromatic C-H modes of the aromatic ring from the terephthalate linker. From Figure 11, the amide sharp peak at 1655 cm^−1^ corresponds to the residue of DMF, which was also observed in other synthesized UiO-66. The terephthalic stretch was clearly observed at 1580 cm^−1^, and 1387 cm^−1^ corresponds to the in and out-of-phase stretching modes of carboxylates. Zr-O modes were observed at 744 and 704 cm^−1^ in Figure 12, while the C=C stretch in the aromatic ring was located at 1504 cm^−1^.

For [EMIM][TFSI], characteristic peak at 3159 cm^−1^ were assigned to H-C-C-H symmetric stretching and NC(H)NC-H stretch. A small peak at 2981 cm^−1^ was the stretching of CH vibrations of the ethyl chain in the [EMIM]^+^ ion. A lot of sharp peaks can be assigned such as 610 cm^−1^ (SOO antisymmetric bending), 1133 cm^−1^ (SOO symmetric stretching), 569 cm^−1^ (CF_3_ antisymmetric bending), 600 cm^−1^ (SOO antisymmetric bending), 650 cm^−1^ (SNS bending) and 741 cm^−1^ (CF_3_ symmetric bending). Meanwhile, a strong peak at 1052 cm^−1^ corresponds to SNS antisymmetric stretching, contributed by ring in-plane antisymmetric stretching, NCH_3_ twisting and C-C stretching. The SOO symmetric stretching was observed at 1132 cm^−1^ and the ring in-plane antisymmetric stretching was identified at 1171 cm^−1^, which was contributed from (N)CH_2_ and CN)CH_3_CN stretch and C-C stretch.

Upon loading of [EMIM][TFSI], several peak shifts and changes in peak intensity were observed, which indicates host–guest molecular interactions. The disappearance of OH broad peaks indicates the dehydroxylated state of IL@UiO-66. The aliphatic CH_2_ stretch at 3159 cm^−1^ was a decrease in intensity, which corresponds to interactions of the ethyl chain of IL with UiO-66. The CH sp^3^ stretches became smaller as they began to overlap with each other.

### 2.3. Density Functional Theory Calculations

The host–guest interactions of UiO-66 and [EMIM][TFSI] were studied by performing a first-principles calculation by solving Schrödinger’s equation without the inclusion of empirical data. Analysis of the interactions of this system was evaluated by comparing the geometry structure of pristine UiO-66 (linker model), pure [EMIM][TFSI] and the combined system of both structures. The optimized structure of these structures is depicted in Figure 13, Figure 14 and Figure 15. The geometry optimization was performed via a self-consistency field, and all structures converged to their respective minimum single-point energies, which were the lowest energy solutions for Schrödinger’s equation; (UiO-66: −1217.233 Eh, IL: −2169.882 Eh, IL@UiO-66: −3387.164 Eh). The binding energy of the following complex was −0.049 Eh, which can be calculated by finding the energy difference in the complex and the sum of energy of UiO-66 and IL.

To visualize the movement of atoms in geometry optimizations of UiO-66 and [EMIM][TFSI], several optimized geometries were compared in Figure 16 until the lowest possible energy was reached. At first, one pair of [EMIM][TFSI] was initially placed between the two organic linkers to mimic the insertion of guest molecules inside the micropores of the UiO-66. Since the oxygen atoms of organic linkers were fixed, the energy of other atoms was minimized so that the IL could interact with the organic linkers. As the SCF iterations proceed, the [EMIM][TFSI] molecule travels towards the oxygen-terminated terephthalate linker. The presence of hydroxyl functional groups increases the strength of intermolecular interactions due to the electron-withdrawing capability of oxygen. The inorganic parts of UiO-66, such as from the zirconium region and inorganic bridge, do not have the capacity for this interaction. Because of the nanocavities of UiO-66, [EMIM][TFSI] was nanoconfined at the organic linker region to create a stable host–guest interaction [41]. Moreover, the anion part of [EMIM][TFSI] has a strong interaction with the organic linker compared to the cation part. The anion contains sulfur and fluorine, which have a similar charge distribution electronegative property as the hydroxyl group from the organic linker, which can provide strong electrostatic forces. The oxygen from the [TFSI]^−^ anion displays strong intermolecular forces with the hydroxyl part of the organic linker, causing an orientation twist with an intermolecular distance of 1.74784 Å, which corresponds to strong hydrogen bonding [42,43].

Several structural parameters of the studied systems were tabulated in Table 2. The changes of the bond distance and bond angle were observed to indicate any possible interactions that occur and to provide information about the strength and nature of the host–guest interactions between the organic linkers and IL. When IL was placed near the UiO-66 linkers, an increase in bond distance occurred between the two bottom hydrogens of terephthalate linkers, suggesting the repulsion of the incoming [EMIM][TFSI]. Angle distortions also occurred at the terminating groups of terephthalate linkers due to the electrostatic potential. The most significant angle distortion was observed at H10-O5-C15 on the organic linkers. The anion part of IL strongly interacts with the hydroxyl group, forcing the bond angle to increase from 110.544° to 114.608°. The electrostatic force between the anion and cation of [EMIM][TFSI] was decreased as they are now preferably interacting with the organic linkers.

In addition to geometry optimizations, another electronic solution of the DFT calculation is molecular orbital, which is a mathematical function that describes the wave-like behavior and location of an electron in a molecule. It was constructed by the linear combination of atomic orbitals, which gives rise to molecular orbitals. Highest occupied molecular orbital (HOMO) and lowest unoccupied molecular orbital (LUMO) are types of molecular orbitals that describe the energy difference of a given material. The prediction of the HOMO-LUMO gap could provide information about the strength and stability of transitional metal complexes and their respective colors in solution [44]. Table 3 represents the value of HOMO and LUMO, while Figure 17 visualizes the energy level diagram of the HOMO-LUMO gap. The HOMO-LUMO energy gap of UiO-66 and IL@UiO-66 was 5.06054 and 5.26054 eV, respectively. As [TFSI]^−^ anion approaches, the electric field of [TFSI]^−^ on the nearby hydrogen atoms alters the electron orbitals. Thus, the electric multipole momentum altered the band gap of the UiO-66 linkers [45,46]. The HOMO and LUMO cloud orbital of UiO-66 was switched when incorporated with IL. Because of the strong interactions of IL and organic linkers, the electronic distribution in the organic linkers was disrupted, which caused the rearrangement of energy levels. In addition, the IL itself has several unique characteristics, such as low vapor pressure and high ionic conductivity, which consequently alters the electronic structure of UiO-66. The energy level was increased when IL was incorporated into the system, which signifies the stability of the system. The enhancement of stability in host–guest interactions is beneficial for the durability of a solid-state electrolyte, which can prevent material degradation and system failure. This can increase the lifespan of an electrochemical device [47]. Furthermore, another aspect of the molecular orbital is the energy level alignment. If the organic linkers and IL have energy that is close to each other, there is a high possibility of molecular interactions between the two species, such as a charge transfer and electron transfer. Based on the molecular orbital energies, the HOMO of UiO-66 was less than the LUMO of IL (−7.57217 eV < −1.21690 eV). This caused electrons to be transferred from IL to organic linkers of UiO-66, resulting in an electron-deficient complex [48,49].

The molecular electrostatic potential map (MESP) can provide more information about the molecular interactions between UiO-66 and IL. Figure 18 shows the MESP map of the three different components of the IL@UiO-66 system. Regions with negative potential (red) show that electrons are easily found, while positive potential (blue) indicates areas where electrons are hardly found. For the organic linkers in UiO-66, large negative potential can be found at the oxygen-terminated terephthalate, and it was hypothesized that the guest molecule can strongly interact in this region. The electron was mainly distributed at the anion site of the IL due to the presence of electronegative atoms such as sulfur, oxygen and fluorine. When IL was present, the electron rich region was distributed at the interface of oxygen-terminated terephthalate and the oxygen from the anion of IL. There was a strong electrostatic attraction between the two components where the electron was transferred from the IL as the electron donor group to the organic linker as the electron accepting group. This is consistent with the energy level alignment of the molecular orbitals of both components. On the other hand, the cation part of the IL interacted less with the organic linker but played an important role by ensuring the stability and structure of the anion [50].

## 3. Methodology

### 3.1. Chemicals

Zirconium(IV) chloride, 98% (ZrCl4) and terephthalic acid, 99+% were purchased from Acros. 1-Ethyl-3-methylimidazolium bis(trifluoromethylsulfonyl)imide, 98% [EMIM][TFSI] was supplied by Sigma Aldrich. Acetic acid (glacial) 100% anhydrous, acetone ACS reagent and N, N-Dimethylformamide (DMF) were purchased from Merck. All chemicals were used without further purification steps.

### 3.2. Preparation of IL@UiO-66

The synthesis of IL@UiO-66 was taken from previous works with few modifications [37,51,52]. First, 1.2 mmol of zirconium(IV) chloride and 1.2 mmol of terephthalic acid were weighed and transferred into a vial. A total of 30 equivalents of acetic acid were poured into the mixture. A total of 30 mL of *N*,*N*-dimethylformamide was pipetted into the vial under fume hood suction. The mixture was swirled, capped and heated at 120 °C in an isothermal oven for 24 h. The solid product (UiO-66) was yielded via centrifugation to remove the solvent, followed by washing with fresh DMF to eliminate unreacted starting materials. The product was then transferred into a petri dish and dried in a vacuum oven at 90 °C for 2 h to remove the leftover solvent. Next, a wet impregnation method was implemented to load the guest molecule onto the UiO-66. A total of 0.35 g of [EMIM][TFSI] was weighed and dissolved in 10 mL of acetone. The mixture was capped and stirred for 15 min. A total of 0.65 g of UiO-66 was weighed and transferred to the mixture and stirred overnight with a closed lid. On the next day, the cap was opened to allow evaporation of acetone under fume hood suction. The sample was dried in a vacuum oven at 80 °C for 2 h to remove the solvent residue. The mixture was hereinafter referred to as IL@UiO-66.

### 3.3. X-ray Diffractions

The crystallography properties of each sample were characterized using Panalytical Xpert3 Powder XRD with Cu Kα radiation (40 kV, 40 mA) at 2θ (2° to 80°) with a scanning rate of 4 °C min^−1^. The diffractogram was collected with 2970 points using a continuous scan mode.

### 3.4. Fourier Transformation Infrared (FTIR) Spectroscopy

Molecular characterizations of samples were performed via Frontier 01 FTIR spectrometer by Perkin Elmer. Before the sample was analyzed, the background spectrum was collected first to eliminate unwanted residue peaks from the sample spectrum. Number of scans was set to 64 with a spectral resolution of 2 cm^−1^.

### 3.5. Scanning Electron Microscopy

Evo LS15 Variable Pressure Scanning Electron Microscopy (VPSEM) was used to characterize the morphology of samples. The images of samples were captured with an acceleration voltage of 10 keV and 10 A acceleration current ranging from 100 to 5 μm.

### 3.6. Brunauer–Emmett–Teller (BET) Surface Area Analysis

The surface area analysis and porosity of materials were conducted via the BET method using the Micromeritic ASAP 2000 instrument where the nitrogen adsorption–desorption curves were recorded.

### 3.7. Computational Method

The periodic structure of UiO-66 was obtained from Crystallography Open Database (entry 4512072) [53]. This structure consists of a zirconium cluster, coordinated with 12 terephthalate linkers, resulting in tetrahedral and octahedral pores. To avoid computational difficulties, the bulk structure of UiO-66 was truncated into two terephthalate linkers. Such linker models were extensively studied for numerous computational calculations [54,55,56]. The coordinates of oxygens were constrained to mimic the rigidity of the UiO-66 periodic structure. Density functional theory calculation was performed on the UiO-66 linkers, [EMIM][TFSI] and the UiO-66-[EMIM][TFSI] complex via the ORCA (5.0.3) package. Becke’s three-parameter exchange with Lee, Yang, and Parr’s (B3LYP) exchange-correlation functional was used for the single point calculation and geometry optimization. The split valence polarization def2-SVP basis set was employed throughout the calculation [57]. The systems were run through a series of self-consistency loops until reaching the minimum energy threshold. The output file was interpreted by Chemcraft to visualize the structural parameters of simulated systems, molecular electrostatic potential and molecular orbitals.

## 4. Conclusions

The crystal properties, morphology, vibrational studies and density functional theory were performed for UiO-66 and IL@UiO-66. The XRD, SEM and EDX mapping confirm the topology and elemental distribution of UiO-66. Aggregation of ILs on the surface of UiO-66 was due to the bunching effect in the micropores of UiO-66. Molecular vibrations revealed in FTIR confirmed the strong interactions of [EMIM][TFSI] with UiO-66, which was also confirmed in the structural parameters study from the density functional theory calculation. Analysis of molecular orbitals of the HOMO and LUMO of both pristine and IL-loaded UiO-66 indicates that the stabilization of material can be achieved by loading guest molecules into the host material. This study will be useful as a guideline to assess other systems of MOF and IL that may have great potential in battery systems.

## Figures and Tables

**Figure 1 molecules-28-02833-f001:**
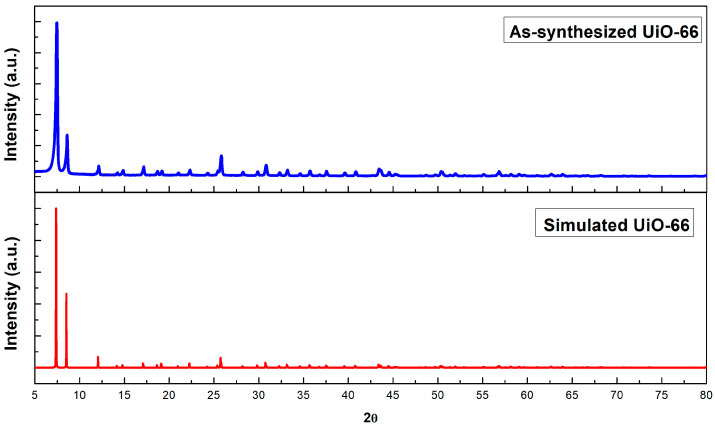
The XRD patterns of as-synthesized UiO-66 and simulated UiO-66.

**Figure 2 molecules-28-02833-f002:**
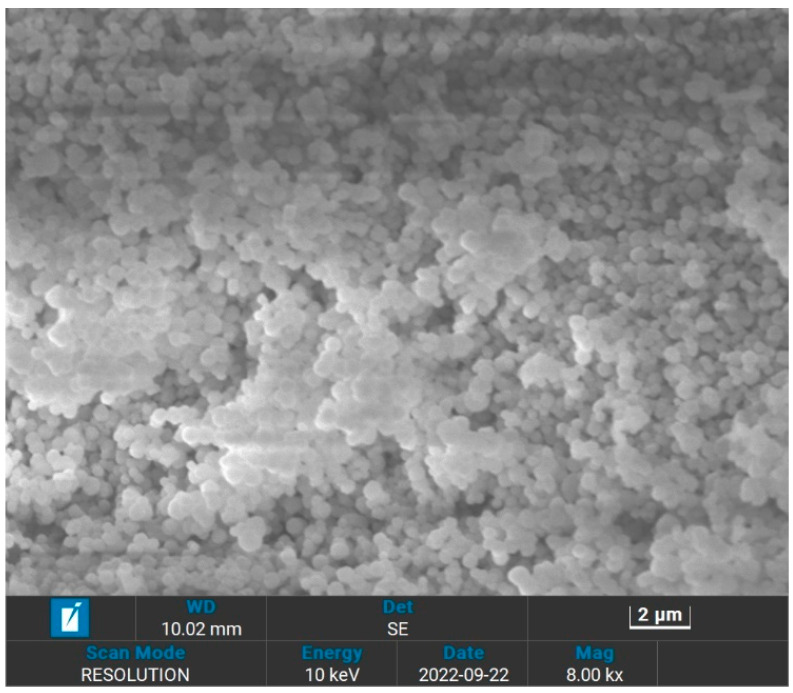
The surface of the pristine UiO-66 at 2 µm with magnification.

**Figure 3 molecules-28-02833-f003:**
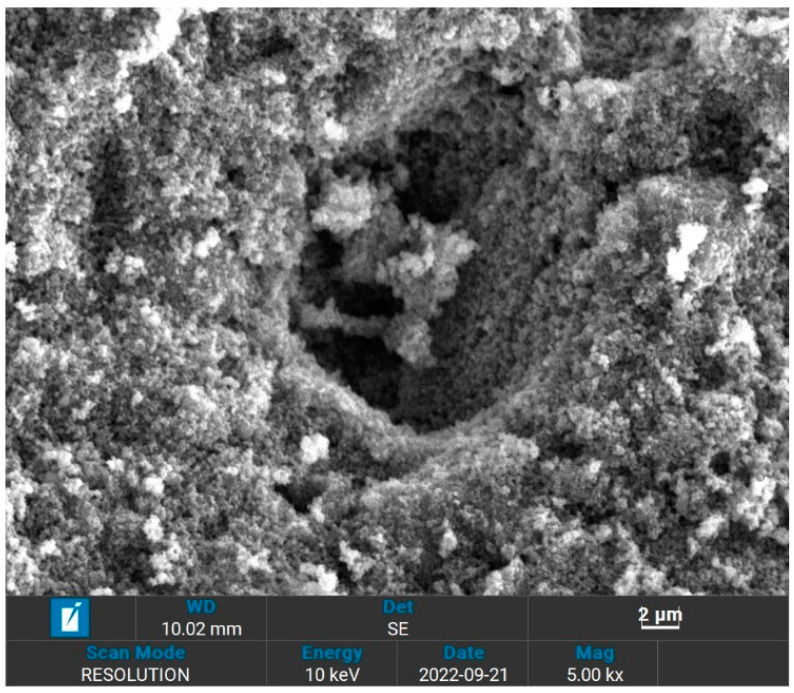
The surface of IL@UiO-66 at 2 µm.

**Figure 4 molecules-28-02833-f004:**
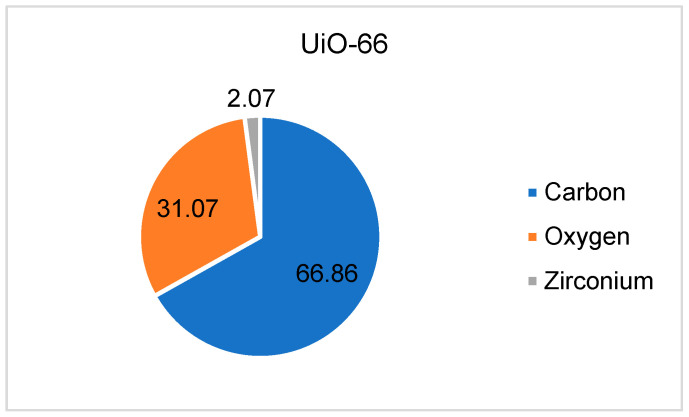
The atomic percentage of UiO-66 calculated from EDX.

**Figure 5 molecules-28-02833-f005:**
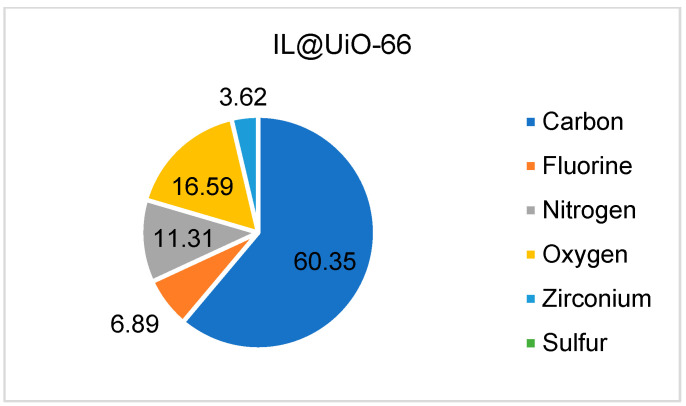
The atomic percentage of IL@UiO-66 calculated from EDX.

**Figure 6 molecules-28-02833-f006:**
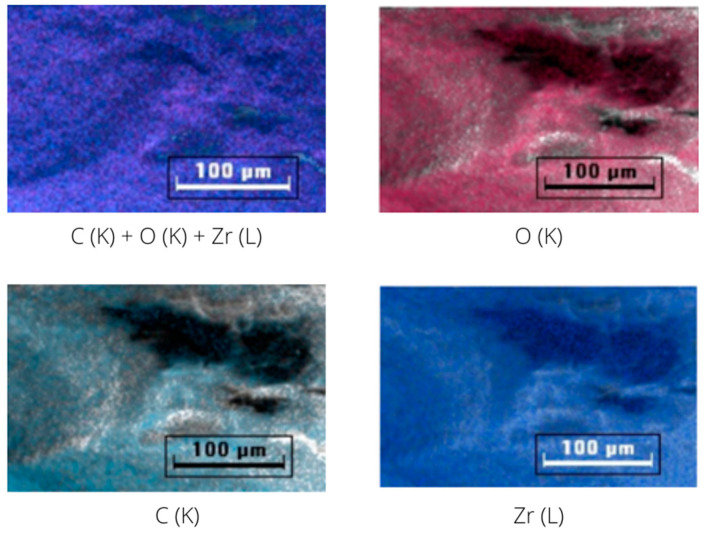
EDX mapping of UiO-66.

**Figure 7 molecules-28-02833-f007:**
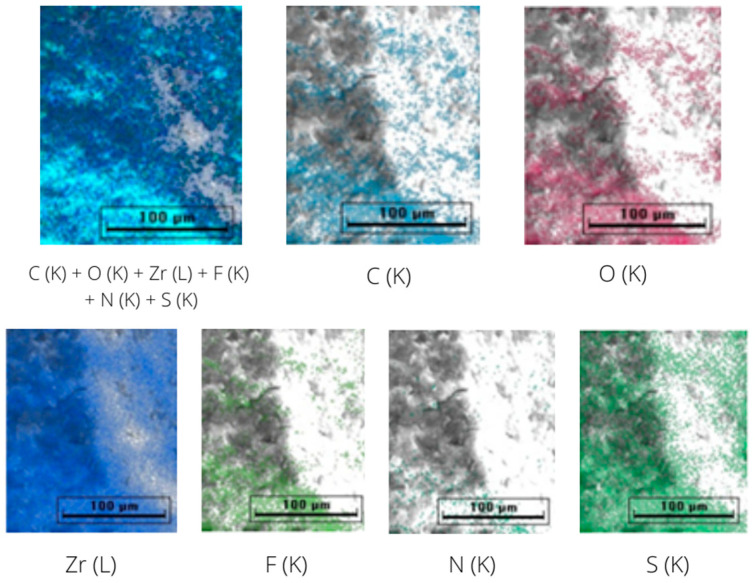
EDX mapping of IL@UiO-66.

**Figure 8 molecules-28-02833-f008:**
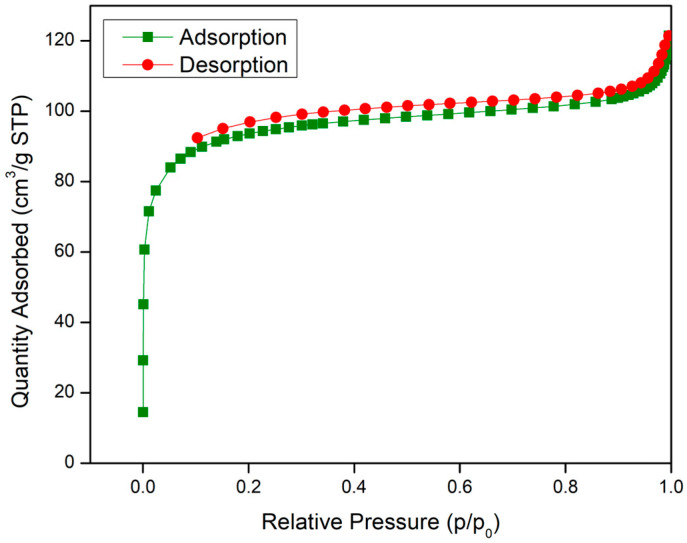
Nitrogen adsorption–desorption isotherms curves for UiO-66.

**Figure 9 molecules-28-02833-f009:**
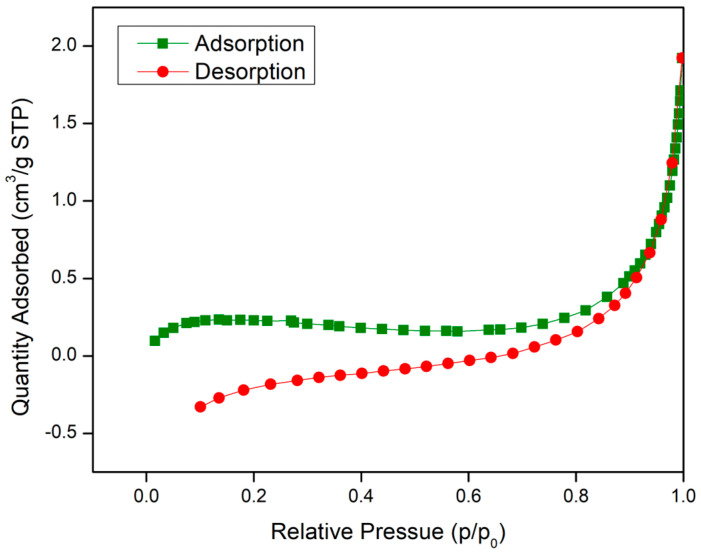
Nitrogen adsorption–desorption isotherms curves for IL@UiO-66.

**Figure 10 molecules-28-02833-f010:**
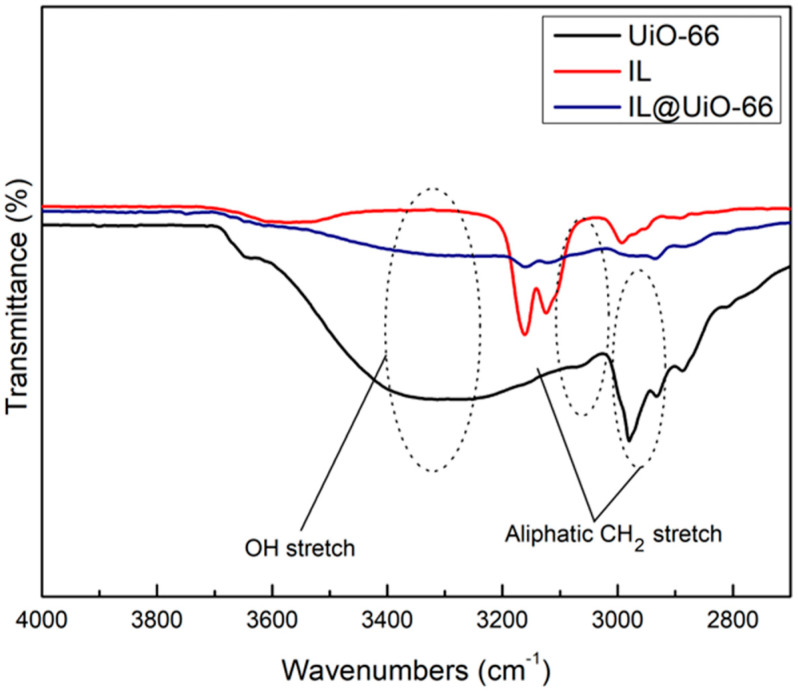
FTIR spectrum of UiO-66, IL and IL@UiO-66 from 4000 to 2700 cm^−1^.

**Figure 11 molecules-28-02833-f011:**
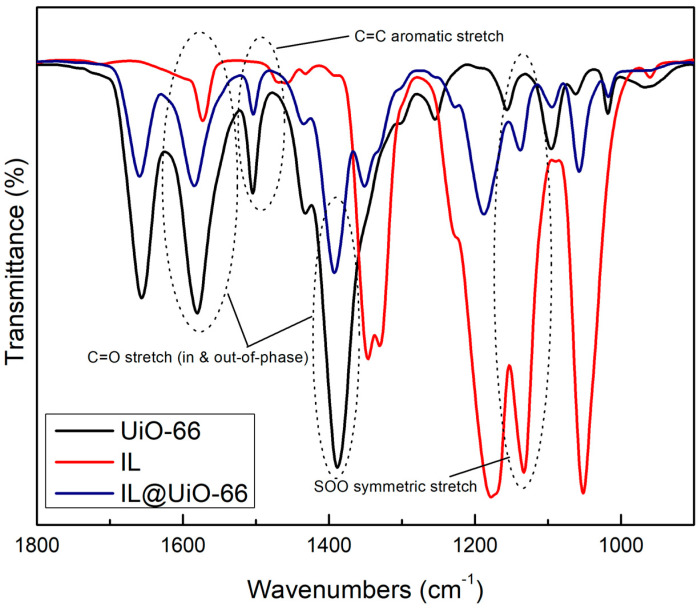
FTIR spectrum of UiO-66, IL and IL@UiO-66 from 1800 to 900 cm^−1^.

**Figure 12 molecules-28-02833-f012:**
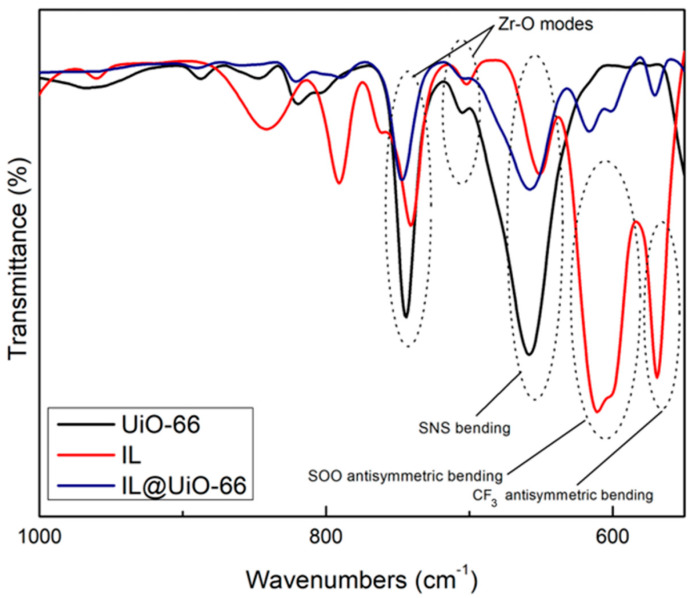
FTIR spectrum of UiO-66, IL and IL@UiO-66 from 1000 to 550 cm^−1^.

**Figure 13 molecules-28-02833-f013:**
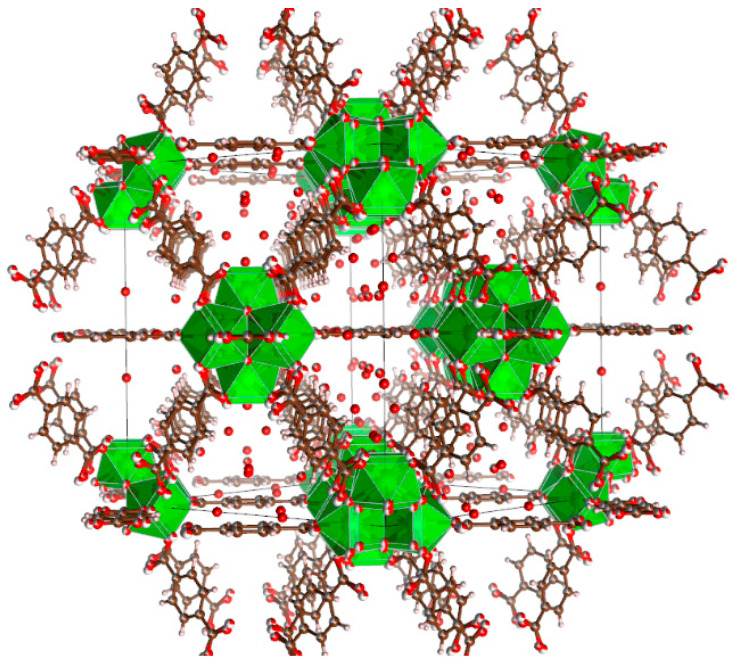
The periodic structure of bulk UiO-66 and the optimized structure of the UiO-66 linker model.

**Figure 14 molecules-28-02833-f014:**
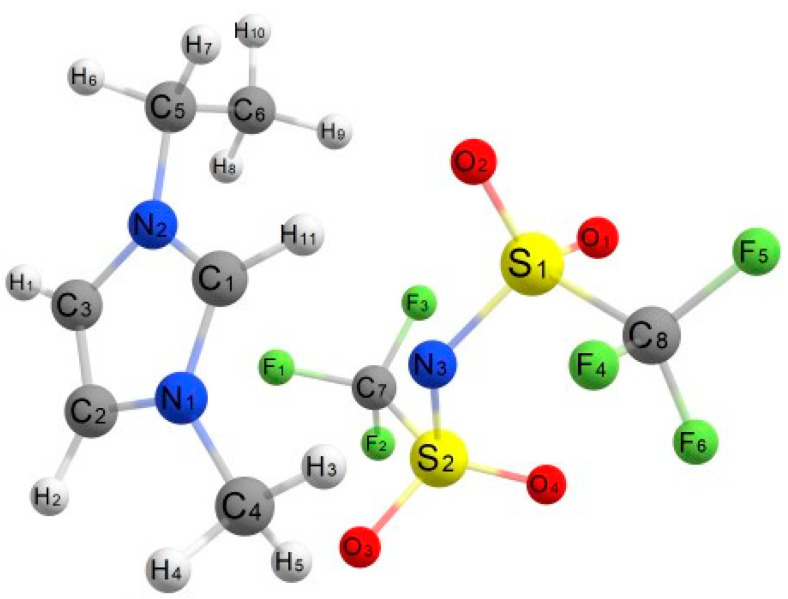
The optimized structure of [EMIM][TFSI].

**Figure 15 molecules-28-02833-f015:**
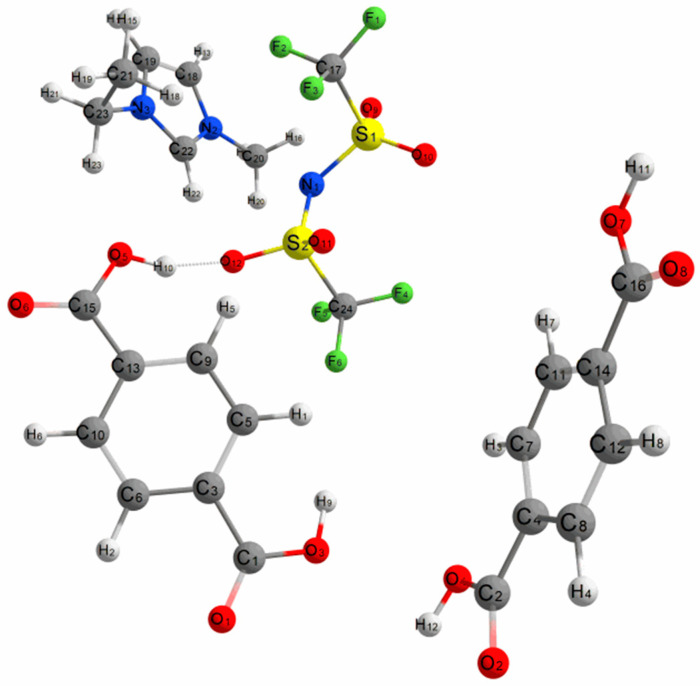
The optimized structure of UiO-66 and [EMIM][TFSI].

**Figure 16 molecules-28-02833-f016:**
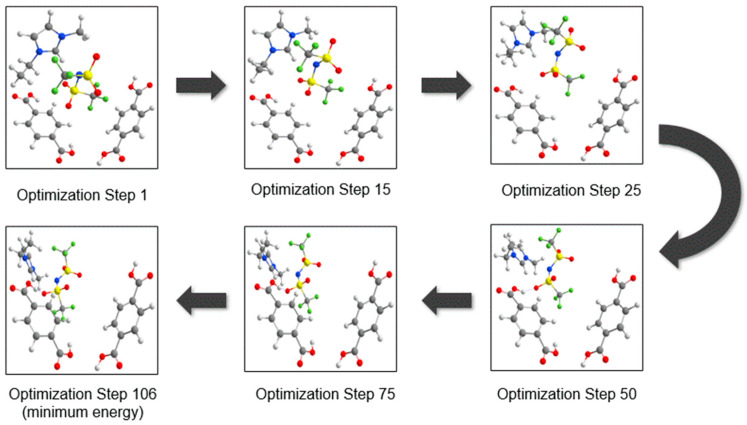
The geometry optimization process of UiO-66 (organic linkers) and [EMIM][TFSI]. The IL was initially placed at the center of the organic linkers, and eventually, the IL was taken to the terminated oxygen.

**Figure 17 molecules-28-02833-f017:**
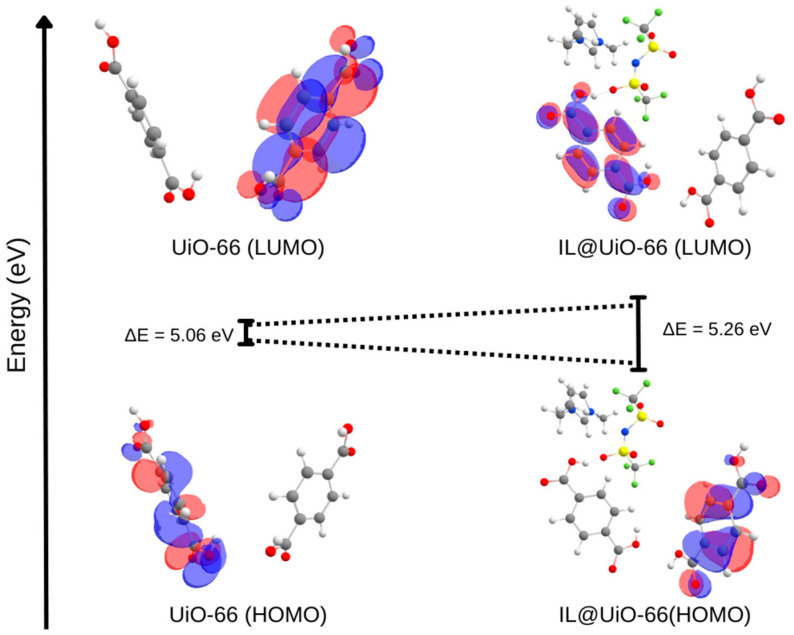
The molecular orbital energy diagram of the highest occupied molecular orbital (HOMO) and lowest unoccupied molecular orbital for UiO-66 linkers and IL@UiO-66.

**Figure 18 molecules-28-02833-f018:**
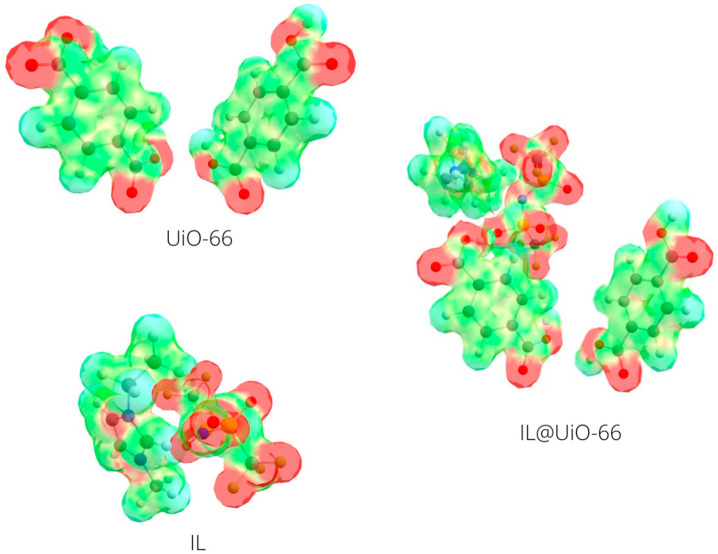
The molecular electrostatic potential isosurface of UiO-66 linkers and [EMIM][TFSI]. Red color represents the attraction of the atom by the concentrated electron density in the molecules, blue color represents the repulsion of the atom by the atomic nuclei in regions where low electron density can be found, and the nuclear charge is incompletely shielded. The contour value was set to 0.1, and the value range was set from −0.1 to 0.05.

**Table 1 molecules-28-02833-t001:** Tabulation of the surface area analysis of UiO-66 and IL@UiO-66.

	BET Surface Area (m^2^/g)	Micropore Area (m^2^/g)	Pore Size (nm)	Pore Volume (cm^3^/g)
UiO-66	295.9922	210.9590	2.45843	0.107151
IL@UiO-66	0.6684	0.7215	15.23378	0.000373

**Table 2 molecules-28-02833-t002:** The structural parameters of UiO-66, IL and IL@UiO-66.

Parameter	UiO-66	IL	IL@UiO-66	Parameter Change
H9-H12	2.78095 Å		3.13221 Å	0.35126
H10-H11	9.88920 Å		9.81681 Å	−0.07239
O7-H10	9.23503 Å		9.25872 Å	0.02369
O4-H9	3.32435 Å		3.13757 Å	−0.18678
Benzene1-Benzene2	7.33452 Å		7.43361 Å	0.09909
H5-C9-C13	121.056°		121.643°	0.587
H7-C11-C14	119.935°		119.809°	−0.126
H10-O5-C15	110.544°		114.608°	4.064
O5-C15-O6	122.942°		122.625°	−0.317
H11-O7-C16	106.966°		106.906°	−0.06
O7-C16-O8	123.319°		122.871°	−0.448
O2 *-H11 *		2.10542 Å	2.08674 Å	−0.01868
O3 *-H5 *		2.25052 Å	2.30103 Å	0.05051
C6 *-F3 *		3.63615 Å	3.55537 Å	−0.08078
C4 *-C8 *		4.52568 Å	4.27397 Å	−0.25171
S1 *-O1 *		1.45839 Å	1.46381 Å	0.00542
S2 *-O4 *		1.45877 Å	1.45747 Å	−0.0013
N1 *-N3 *		3.09522 Å	3.32384 Å	0.22862
N2 *-N3 *		3.57987 Å	4.09536 Å	0.51549
N3 *-H11 *		2.62692 Å	2.65812 Å	0.0312

The atoms for the IL part were indicated with asterisk (*).

**Table 3 molecules-28-02833-t003:** The value of the highest occupied molecular orbital (HOMO) and lowest occupied molecular orbital (LUMO) for UiO-66, IL and IL@UiO-66.

	UiO-66	IL	IL@UiO-66
HOMO (eV)	−7.57217	−7.11229	−7.21569
LUMO (eV)	−2.51163	−1.21690	−1.95515
(LUMO-HOMO) (eV)	5.06054	5.89539	5.26054

## Data Availability

Not applicable.

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
