# Peer review of "Host–Guest Interactions of Zirconium-Based Metal–Organic Framework with Ionic Liquid"

_molecules, 2023, doi:10.3390/molecules28062833_

Round 1
Reviewer 1 Report
In this paper, the authors investigated the host-guest interaction of IL and MOF, which could be a promising candidate as electrolyte material in energy storage system. Overall, this work has performed well in detailed experimental studies and has important implications for the study of MOFs in the context of energy materials. I would like to recommend it for publication in Molecules after the following point can be well addressed.
1. The host and guest symbols are incorrectly marked, usually with a mark like (guest)@(host), and should be modified by the authors.
2. The resolution of the figures are too low, the author should upload high resolution figures
3. The author lists too many redundant images, for example of UiO-66 at different magnifications (Figure 2-4 and Figure 5-7), which should be streamlined or combined. Only the important images need to be listed in the text, and the rest can be placed in the supplementary material.
4. Why did the authors not include the Zr cluster when performing DFT calculations to model the optimized structure of UiO-66?
5. In MOF, the ligand terephthalic acid has been deprotonated and there is no hydrogen on its carboxyl group, but in Figure 17 the authors depict an interaction between the host and guest through hydrogen bonding that needs to be further interpreted.
6. Some errors in the manuscript, e.g. line181 “UiO-@IL”.
7. It is recommended to add recent articles on design strategies and applications of MOF materials. (e.g., 10.1021/jacs.1c11750, 10.1002/anie.202105830).
Author Response
In this paper, the authors investigated the host-guest interaction of IL and MOF, which could be a promising candidate as electrolyte material in energy storage system. Overall, this work has performed well in detailed experimental studies and has important implications for the study of MOFs in the context of energy materials. I would like to recommend it for publication in Molecules after the following point can be well addressed.
Response : We thank reviewer for the insightful comment. We have address all comments and suggestion from the reviewer.
- The host and guest symbols are incorrectly marked, usually with a mark like (guest)@(host), and should be modified by the authors.
Response: The reviewer raised concern about the format of host and guest symbol. The correct form should be IL@UiO-66. We have revise the format in this manuscript.
- The resolution of the figures are too low, the author should upload high resolution figures.
Response: The reviewer raised concern about the low quality of figures. We acknowledge the feedback from reviewer and we have try our best to maximize the quality of the figure. We have revised the XRD graph (Figure 1), FTIR spectrum (Figure 14, 15 and 16), the optimized structure of IL@UiO-66 (Figure 19) and the geometry optimization steps (Figure 20). Some of the figures are already in its best quality due to the limitation of the instrument.
- The author lists too many redundant images, for example of UiO-66 at different magnifications (Figure 2-4 and Figure 5-7), which should be streamlined or combined. Only the important images need to be listed in the text, and the rest can be placed in the supplementary material.
Response 4: The reviewer stated the there are too many redundant image of SEM image of UiO-66. We agree that only the important iamge should be placed in the manuscript. Therefore, we have select the SEM image of UiO-66 and IL@UiO-66 at 2 µm, while the rest of the images are included in a separate Supplementary Informations.
- Why did the authors not include the Zr cluster when performing DFT calculations to model the optimized structure of UiO-66?
Response: The reviewer raised concern about the exclusion of zirconium cluster in this study. There are many ways to make a DFT model for metal-organic frameworks such as bulk MOF, full linker sterics, catalytic pocket, linker model and linker excitation transfer to node. Because of its supramolecular structure behavior, the inclusion of metal cluster will make the calculation more expensive. Besides, the UiO-66 is more complex than other simpler MOF which requires more time to calculate. Therefore, we try to use the linker model by truncating the terephthalic acid linkers from the CIF file of UiO-66 to mimick the real structure of UiO-66.
- In MOF, the ligand terephthalic acid has been deprotonated and there is no hydrogen on its carboxyl group, but in Figure 17 the authors depict an interaction between the host and guest through hydrogen bonding that needs to be further interpreted.
Response: The reviewer raised concern about the present of hydrogen in our DFT model. When truncating the linkers from the bulk UiO-66, the carboxyl part need to be terminated with hydrogen the ensure the charge neutrality of the molecule. Before this, we have try to not include the hydrogen, however, the optimization does not achieve convergency. We suspect that this might be due to the unbalance charge in our structure.
- Some errors in the manuscript, e.g. line181 “UiO-@IL”.
Response: The reviewer raised concern about the error in line 181. We have now corrected it to IL@UiO-66.
- It is recommended to add recent articles on design strategies and applications of MOF materials. (e.g., 10.1021/jacs.1c11750, 10.1002/anie.202105830).
Response: The reviewer suggestes to include recent citation of application of MOF materials. We have now included the article suggested by the reviewer in our introduction part.
Reviewer 2 Report
1. In Figure 1, the quality of this is poor, also pls integrate this two part into one map.
2. Fig3-Fig7 could be combined with one figure, such as a. b, c….
3. Please provide the BET for the UIO-66 and UiO-66@IL
4. Mark the main peaks in Fig. 13.
5. “The possible intermolecular forces in MOF are van der Waals forces, π-π interactions, hydrogen bonding and stabilization of π-bond by polarized bonds from the synergistic interaction of metal and ligands.” This should be cited the refs, such as Inorganics, 10(2022) 202 and Micropor. Mesopor. Mat, 341(2022) 112098. “The geometry of MOF is adjustable depending on the type of metal nodes and length and functional groups of organic ligands.” It should be added the documents, such as Mater. Today. Commum., 2022, 31,103514; J. Solid State Chem. 318(2023) 123713 and J. Mater. Chem. B., 2022, 10, 5105 – 5128.
6. The quality of Fig. 17/18 is also poor, pls improved it.
7. Give the possible molecular interactions for this synthesized process.
8. The manuscript contains spelling/grammatical errors. So, the language should be polished thoroughly.
Author Response
- In Figure 1, the quality of this is poor, also pls integrate this two part into one map.
Response: The reviewer raised concern on the poor quality of XRD diffractogram and suggest to integrate both part into one graph. We acknowledge the reviewer and we have now revise the quality of XRD and integrate both part in one graph.
- Fig3-Fig7 could be combined with one figure, such as a. b, c….
Response: The reviewer recommended to combine the SEM figures. As suggested by Reviewer 1, we have select the important SEM image to be included in the main manuscript. The other SEM images are presented in the Supplementary Information document.
- Please provide the BET for the UIO-66 and UiO-66@IL.
Response: The reviewer suggested to provide the BET for UiO-66 and IL@UiO-66. We have analyzed the BET surface are and discuss the result in the result discussion.
- Mark the main peaks in Fig. 13.
Response 4: The reviewer suggested to mark the main peaks of FTIR spectrum. We have now revise the FTIR spectrum with marked main peaks (Figure 14, 15 and 16).
- “The possible intermolecular forces in MOF are van der Waals forces, π-π interactions, hydrogen bonding and stabilization of π-bond by polarized bonds from the synergistic interaction of metal and ligands.” This should be cited the refs, such as Inorganics, 10(2022) 202 and Micropor. Mesopor. Mat, 341(2022) 112098. “The geometry of MOF is adjustable depending on the type of metal nodes and length and functional groups of organic ligands.” It should be added the documents, such as Mater. Today. Commum., 2022, 31,103514; J. Solid State Chem. 318(2023) 123713 and J. Mater. Chem. B., 2022, 10, 5105 – 5128.
Response: The reviewer suggested to include the recommended references in our introduction part. We have now cite the references provided by the reviewer.
- The quality of Fig. 17/18 is also poor, pls improved it.
Response: The reviewer raised concern about the low quality images in Figure 17 and 18. We have now revised the illustration with higher quality image (the figures are now referred Figure 19 and 20 due to the addition of figures in previous section).
- Give the possible molecular interactions for this synthesized process.
Response: The reviewer suggested to put the possible molecular interactions of the as-synthesized UiO-66 and IL@UiO-66. We have now included the discussion in the first part of our discussion section:
Coordination of Zr4+ and terephthalic acid lead to the self-assembly of metal-organic clusters. The clusters will grow in size and forms nuclei, which were the seed of the UiO-66 crystal. When more clusters were formed, the crystal grow in size and reach its final form. To form IL@UiO-66, IL was diffused into the pores of the UiO-66 structure via wet-impregnation method using acetone, and it will interact with the metal nodes and organic linkers via host-guest interactions. Possible molecular interactions are coordination interactions of electronegative atom from IL (nitrogen or oxygen) with zirconium node, electrostatic forces between cation and anion and weak van der Waals interaction.
- The manuscript contains spelling/grammatical errors. So, the language should be polished thoroughly.
Response: The reviewer raised concern about the spelling and grammatical errors in our manuscript. We have now revised the spelling and grammatical errors in this manuscript.
Round 2
Reviewer 1 Report
My comments were addressed satisfactorily. The manuscript is much improved, and I'd be happy to see it published.
Reviewer 2 Report
The authors have replied my comment, but the refs formation is not followed the journal rules. Pls check and revise when it was in proof.